# A Strategy of In Situ Catalysis and Nucleation of Biocompatible Zinc Salts of Amino Acids towards Poly(l-lactide) with Enhanced Crystallization Rate

**DOI:** 10.3390/polym11050790

**Published:** 2019-05-02

**Authors:** Yuan Liang, Meili Sui, Maomao He, Zhiyong Wei, Wanxi Zhang

**Affiliations:** 1School of Materials Science and Engineering, Jilin University, Changchun 130022, China; lyhoneybook@hotmail.com (Y.L.); zhangwanxi0626@sina.com (W.Z.); 2Department of Polymer Science and Engineering, School of Chemical Engineering, Dalian University of Technology, Dalian 116024, China; ml_sui68206@163.com (M.S.); hemao@mail.dlut.edu.cn (M.H.)

**Keywords:** zinc salts of amino acids, poly(l-lactide), catalysis and nucleation

## Abstract

The intrinsic drawback of slow crystallization rate of poly(l-lactide) (PLLA) inevitably deteriorates its final properties of the molded articles. In this work, we proposed a new strategy towards poly(l-lactide) with enhanced crystallization rate by ring opening polymerization (ROP) of l-lactide (l-LA) catalyzed by biocompatible zinc salts of amino acids. For the first time we developed a one-pot facile method of zinc salts of amino acids acting dual roles of catalysis of l-LA polymerization and in situ nucleation of the as-prepared PLLA. Nine zinc salts of different amino acids, including three kinds of amino acids ligands (alanine, phenylalanine, and proline) with l/d-enantiomers and their equimolar racemic mixtures, were first prepared and tested as catalysts of l-LA polymerization. A partial racemization was observed for zinc salts of amino acids whereas no racemization was detected for the reference stannous octoate. The polymerization mechanism study showed that the interaction of zinc salts of amino acids and benzyl alcohol forms the actual initiator for l-LA polymerization. Isothermal crystallization kinetics analysis showed that the residual zinc salts of amino acids exhibited a significant nucleation effect on PLLA, evidenced by the promotion of the crystallization rate, depending on the amino acid ligand and its configuration. Meanwhile, the residual zinc salts of amino acids did not compromise the thermal stability of the pristine PLLA.

## 1. Introduction

Poly(l-lactic acid) or poly(l-lactide) (PLLA) is one of the most important biobased and biodegradable polymers, due to its outstanding properties such as biodegradability, biocompatibility, and renewability [1]. As a consequence, PLLA has been widely used as drug carrier, tissue engineering, and food packaging [2]. Ring opening polymerization (ROP) is the most preferable method for the synthesis of PLLA with controlled molecular weight and narrow polydisperisty index [3]. Over the past decades, a large number of metal derivatives have been widely investigated as catalysts for ROP of l-lactide by numerous researchers [4]. However, some catalysts exhibit cytotoxicity associated with both the metals and the ligands, of which are not easily removed from the polymer. Even for tin(II) 2-ethylhexanoate or stannous octoate (Sn(Oct)_2_), the commonly used catalysts for the production of PLLA [5,6,7,8], the safety is at best questionable in the case of food packaging and biomedical materials [4]. Therefore, it is desirable to develop biocompatible or nontoxic metal catalysts for the ROP of l-lactide to produce biocompatible PLLA [4]. 

Zinc compounds are ideal alternative catalysts as they are nontoxic and zinc participates in human metabolism so one can envision their use for the synthesis of biocompatible PLLA for biomedical fields. Some zinc metal and zinc compounds have been firstly investigated by several research groups with mixed results [4]. Organozinc compounds, such as zinc lactate, zinc octanoate, zinc stearate, zinc acetate, and zinc acetylacetonate [9,10,11,12,13,14], were also investigated as catalysts in ROP of l-lactide. On the contrary, many Zn complexes supported by a variety of ancillary ligands have shown as highly active and isoselctive catalysts [15,16,17], but the toxicity of these ligands, which required complicated synthetic process, is unknown [11]. Recently, some natural amino acids as the ligands for zinc complexes [17,18,19] arrested our attention for its nontoxic nature and low cost. Furthermore, amino acids easily reacted with zinc or zinc acetate forming zinc salts of amino acids, which are potential catalysts for ROP of lactides and lactones [11]. However, zinc salts of amino acids have rarely been used as catalysts for the polymerization of l-lactide [20,21,22]. 

Nowadays, PLLA has raised interest as a potential replacement for petroleum-based polymers not just biomedical but also commodity applications due to its good mechanical properties. Nevertheless, the widespread use of PLLA is blocked by some of its intrinsic drawbacks, for example, the slow crystallization rate results in a low degree of crystallinity for PLA materials obtained during an injection-molding cycle involving a high cooling rate and a relatively low mold temperature [23,24]. Addition of nucleating agent (NA) into the polymer matrix is one of the most common methods to improve its crystallization rate [25,26]. A conventional strategy for NA addition should be composed of a two-step process of chemical polymerization and physical blending with NA. This can be complex, costly, and time-consuming. For melt processing of PLLA, some zinc metal salts, such as zinc phenylphosphonate [27,28,29,30], zinc citrate [31], and zinc phenylmalonate [32], were found to be the most effective nucleating agents. We surmised from our experience that the aforementioned zinc salts of amino acids could be considered as potential nucleating agents towards the enhancement of crystallization rate of PLLA. 

In the present work, we focused on the development of a novel in situ polymerization of l-lactide in the presence of zinc salts of amino acids to maximize the potential of in situ nucleation as an ideal strategy [33]. This novel strategy resulted in simultaneous l-lactide polymerization and nucleation in the presence of zinc amino acids. That is, zinc salts of amino acids first catalyze the ring opening polymerization of l-lactide, then the residual zinc salts of amino acids act as a second role of NA accelerating the crystallization of PLLA. The benefit of this strategy could save significant time and cost due to the one-step process of polymerization and nucleation [33].

More specifically, in this work, we performed in situ polymerization of l-lactide using biocompatible catalysts zinc salts with a variety of amino acids and coinitiator benzyl alcohol in bulk, and evaluated the crystallization kinetics of as-prepared PLLA without removal of catalysts. Our current study develops a green and low-cost approach of zinc salts of amino acids combining catalysis and nucleation dual roles, for the production of biocompatible PLLA suitable for broad commodity and engineering plastics by overwhelming the drawback of the poor crystallization ability of PLLA.

## 2. Materials and Methods

### 2.1. Materials

Nine amino acids: l-alanine (l-Ala, 99%), d-alanine (d-Ala, 99%), dl-alanine (dl-Ala, 99%), l-phenylalanine (l-Phe, 99%), d-phenylalanine (d-Phe, 99%), dl-phenylalanine (dl-Phe, 99%), l-proline (l-Pro, 99%), d-proline (d-Pro, 99%), dl-proline (dl-Pro, 99%), were purchased from J&K Chemical Co., Ltd. Benzyl alcohol (BnOH, 99%), zinc acetate (Zn(OAc)_2_, 99%), and stannous octoate (Sn(Oct)_2_, 95%) were obtained from J&K Chemical Co., Ltd. l-lactide (>99%) was kindly provided by Jiangsu SUPLA Bioplastics Co. Ltd. (Suqian, China). Methanol was supplied by Tianda Chemical Reagent Factory (Tianjin, China).

### 2.2. Synthesis of Zinc Amino Acids (Zn(AA)_2_)

We take the synthesis of zinc l-alaninate (Zn(l-Ala)_2_) as an example: Zn(OAc)_2_ (25 mmol) and l-alanine (50 mmol) were suspended in methanol (100 mL) and refluxed for 5 h. The precipitated product was filtered off from the methanol solution and washed by water. The white powder was dried at 40 °C for 48 h prior to use. 

Elementary analysis: Calc. C 29.87 H 4.97 N 11.61; Found C 29.65 H 4.84 N 11.52. 

Other zinc amino acids were prepared by using similar procedure. All FTIR spectra were shown in Appendix A.

### 2.3. Preparation of PLLA Catalyzed by Zinc Amino Acids (Zn(AA)_2_)

PLLA samples were synthesized by different zinc amino acids (28.8 mg, fixed 1 wt % of l-LA) catalyzed polymerization of l-lactide (l-LA, 2.88 g, 20 mmol) using benzyl alcohol (BnOH, 21.6 mg, 0.2 mmol) as initiator with l-LA/BnOH ratio of 100/1 at 130 °C for 24 h in bulk under nitrogen. The as-prepared polymers without any purification were subjected to analysis unless otherwise noted. PLLA was obtained by Sn(Oct)_2_ catalyzed ROP of l-LA under the same conditions as a reference.

### 2.4. Characterization

The molar mass and dispersity (*Ð*) were measured by size-exclusion chromatography (SEC) using a Waters HPLC system, which was equipped with 2414 refractive index detector, 1515 pump, HT 5 Styragel^®^, and Shodex SM-105 standards PS with molecular weights from 1.2 × 10^3^ to 2.75 × 10^6^ g/mol. The tetrahydrofuran (THF) was used as mobile phase at a rate of 1 mL/min and the column temperature of 30 °C.

Fourier transform infrared (FTIR) spectroscopy was performed at ambient temperature on a Thermo Nicolet-20DXB spectrometer. The spectra were recorded in the wavenumber range of 400–4000 cm^−1^ with a resolution of 2 cm^−1^. Zinc amino acids were mixed with potassium bromide (KBr) and pressed into flakes for FTIR measurements.

The specific optical rotation ([*α*]) of the PLLA samples was measured in chloroform at a concentration of 10 g/L and 25 °C using a Jasco P-1010 polarimeter (JASCO Corporation, Tokyo, Janpan) at a wavelength of 589 nm. JASCO Corporation

^1^H NMR measurements were performed on Bruker Avance II 400 MHz spectrometer (Bruker, Karlsruhe, Germany) with deuterated chloroform or trifluoroacetic acid and tetramethylsilane as solvent and reference, respectively.

Matrix-assisted laser desorption ionization time-of-flight mass spectrometry (MALDI-TOF MS) was performed using a Waters MALDI micro MX. One hundred shots were accumulated for the spectra at a 25 kV acceleration voltage in the reflector mode and calibrated using polystyrene as the internal standard.

Isothermal crystallization kinetics was carried out on a Mettler-Toledo DSC1 in aluminum pans under nitrogen atmosphere. The samples (ca. 6 mg) were initially melted at 200 °C for 3 min, and then rapidly cooled to 125 °C at −60 °C/min, the heat flows were recorded for kinetics analysis. 

Thermogravimetric analysis was performed by Q500 (TA instruments, Newcastle, DE, USA) from 100 to 500 °C at a heating rate of 20 °C/min under nitrogen atmosphere with a flow rate of 50 mL/min. 

## 3. Results and Discussion

### 3.1. Synthesis and Characterization

As shown in Scheme 1, nine zinc salts of different amino acids (Zn(AA)_2_), that is, three kinds of amino acids ligands (alanine, phenylalanine, and proline) with two configurations and their equimolar racemic mixtures, were first prepared and then tested as potential catalysts of ROP of l-lactide. The evaluation of them as in situ nucleating agents for the resulting PLLA will be discussed in next section. For the sake for consistent evaluation of catalysis and nucleation, a series of polymerizations were carried out in bulk at 130 °C for 24 h at fixed l-LA/BnOH ratio of 100/1 with high concentration of 1 wt % Zn(AA)_2_. 

As shown in Table 1, the apparent molar masses measured by SEC were consistent with the theoretical values calculated from the designed ratio of monomer to initiator, however, it should be noted that SEC measurements calibrated with polystyrene overestimate systematically the real molecular weights of aliphatic polyesters because of different hydrodynamic volume. Nevertheless, it suggested that the molar mass could be well controlled by the feed ratio. Interestingly, the PLLA obtained by Sn(Oct)_2_ has 100% optical purity without any racemization from l-LA, however, the optical rotations of PLLAs obtained by zinc amino acids were remarkably reduced depending on the kind of amino acids, indicating the occurrence of a partial racemization. The proline affects less strongly the optical purity than other amino acids in zinc salt of amino acids. This result was in agreement with the previous result reported by Kricheldof [11], who attributed it to the basic nature of zinc salts of amino acids with a racemization mechanism.

For better understanding into the polymerization characteristics and final composition of the product, a PLLA oligomer was synthesized at low ratio of l-LA/BnOH to 10. The product was quenched with cold ethanol and the residue recovered was subjected to ^1^H NMR spectroscopy and MALDI-TOF. Figure 1 revealed the presence of benzyl ester end-groups appearing at δ 7.3 ~ 7.4 ppm, which clearly demonstrated that BnOH has an active role as the initiator of ROP of l-LA. This result showed that the energy of activation of initiation involving alcohols is much lower than that of initiation by sole zinc amino acids. In addition to benzyl ester end-groups, CH-OH end groups were found at δ 4.35 ppm. The integral area ratio of peak a and peak d′ from Figure 1 was 4.7:1, that is, the molar ratio was 0.94:1, almost identical quantities. Furthermore, deducted from the integral area of the CH_2_ of the benzyl group overlapping at 5.15 ppm, the integral area ratio of peak d and peak d′ from Figure 1 was 17:1, suggesting that the value of n was almost 9, which was closed to the feed ratio of 10. It was also supported by the high peak among of a series of peaks from the analysis of MALDI-TOF MS, as shown in Figure 2. In addition, these peaks also perfectly supported that the molecular chains of PLLA contain BnO-residue and hydroxyl chain end groups. This result also confirmed that BnOH is the actual initiator of Zn(D-Phe)_2_-catalyzed ROP of l-LA. 

There are two main mechanisms for the ROP of lactide—the coordination insertion mechanism for metal complexes and the activated monomer mechanism for organo/cationic initiators [4,5,6,7,8,12,13,14]. In an attempt to understand better the nature of initiating species of zinc amino acids catalyzed ROP of l-LA in the presence of BnOH, we took Zn(l-Ala)_2_ as an example to mix equimolar amount of monomer and initiator in solution, respectively, then examined the mixtures by ^1^H NMR to identify what the catalyst reacted with is monomer or initiator. 

As shown in Figure 3, the chemical shifts of l-LA monomer did not show any changes after the addition of Zn(l-Ala)_2_ catalyst, whereas BnOH demonstrated an downfield shift under the same condition. It provided a direct proof that the catalyst first reacted with the initiator BnOH then initiated the monomer l-LA. We can speculate that Zn(AA)_2_ has a first interaction with BnOH. As a result, it is proposed that an intermediate product of Zn(AA)_2_ with BnOH is the actual initiator, proceeded by the activated-alcohol pathway not the nucleophilic-attack pathway. Similar results have been previously observed with Sn, Zn, and other metal alkoxides [4,5,6,7,8,12,13,14]. 

### 3.2. Isothermal Crystallization Kinetics

To investigate the in situ nucleation effect of the residual zinc amino acids (Zn(AA)_2_) on the synthesized PLLA listed in Table 1, all of the pristine polymers without any purification were subjected to the DSC analysis. In this regard, isothermal crystallization was to evaluate the crystallization rate of PLLAs containing residual catalysts. 

Figure 4 showed the DSC curves of three series of PLLAs crystallized at 125 °C. As a comparison, the exotherm of PLLA obtained by Sn(Oct)_2_ was also drawn in the Figure 4. Obviously, the exothermal peak of PLLA obtained by Sn(Oct)_2_ is wide, while the exothermal peaks of PLLA obtained by Zn(AA)_2_ sharpen and vary depending on ligand and its configuration. That is, all the PLLAs obtained by Zn(AA)_2_ showed faster crystallization rate than that of PLLA obtained Sn(Oct)_2_. In particular, Zn(l-Ala)_2_ catalyzed PLLA displayed the slowest crystallization rate, Zn(l-Pro)_2_ catalyzed PLLA showed the fastest crystallization rate, and other Zn(AA)_2_ catalyzed PLLAs showed a moderate crystallization rate. The results demonstrated that the residual zinc amino acids showed a significant nucleation effect on PLLA. 

Furthermore, the relative crystallinity (*X*_t_) at a given crystallization time (*t*) can be calculated from the integrated area of the DSC curve from *t* = 0 to *t* divided by the whole area of the exothermal peak. The conversion curves of *X*_t_ versus *t* for PLLA obtained by Zn(Pro)_2_ and Sn(Oct)_2_ crystallized at 125 °C were presented in Figure 5a, others were shown in Appendix A. It was found that the optically pure PLLA obtained by Sn(Oct)_2_ completed the crystallization within 20 min. In contrast, the residual Zn(l-Pro)_2_ can significantly shorten the crystallization time of PLLA up to 2 min.

The Avrami equation is the most common method to analyze the isothermal crystallization kinetics of polymers, and its logarithmic form can be expressed as ln[−ln(1 − *X*_t_)] = ln*k* + *n*ln*t*, where *k* is the crystallization rate constant, and *n* is the Avrami exponent, which is dependent on the nature of nucleation and the dimension of crystal growth [31,34]. The parameters *n* and *k* can be obtained from the slopes and intercepts of fitted lines, respectively, by plotting ln[−ln(1 − *X*_t_)] against ln*t*. Figure 5b showed such plots for the samples isothermally crystallized at 125 °C, and others were shown in Appendix A. The values of *n*, *k* and crystallization half-time (*t*_1/2_) are listed in Table 2. The values of *n* fluctuated within the range of between 2.35 to 2.92, almost irrespective of the kind of zinc amino acids, except for 3.68 from Zn(l-Phe)_2_, suggesting a three-dimensional spherulite growth model with heterogeneous nucleation. According to the data of *t*_1/2_, the nucleation ability of Zn(AA)_2_ with proline ligand is stronger than that of two other series of Zn(AA)_2_ with alanine and phenylalanine ligands, regardless of configuration. The nucleation activity may be attributed to the epitaxial nucleation model based on the lattice matching relationship between crystal structures of PLLA crystals and zinc salts of amino acids.

In addition, we should take into account the effect of optical purity of the resulting PLLA catalyzed by various catalysts on its crystallization rate. Our previous results [35] demonstrated that the crystallization rate of PLLA largely depends on its optical purity. The higher optical purity, the faster crystallization rate. The PLLA obtained by Sn(Oct)_2_ can complete the crystallization within 20 min due to its highest 100% optical purity. Therefore, given the lower optical purity of PLLA obtained by Zn(AA)_2_, the Zn(AA)_2_ catalysts exhibit much stronger nucleation effect than Sn(Oct)_2_. However, for each salt of zinc amino acid, the effect of optical purity of crystallization rate overlapped with its nucleation effect, and they have a positive correlation. As a result, it is difficult to compare the nucleation effect of each zinc salt of amino acid. The comparative study of the sole nucleation effect of zinc salts of amino acids on the identical PLLA matrix is in progress, and the detailed nucleation mechanism of zinc salts of amino acids will be explained in the next article of this series [36].

### 3.3. Thermal Stability 

Thermal stability is considered as one of the most important performance of polymer materials. Especially, PLLA usually starts thermal decomposition reactions at temperature above 200 °C, resulting in the reduction of molecular weight and mechanical properties [37]. Many researches demonstrated that the residual metallic catalysts in PLLA induced the thermal degradation [38,39]. Clearly, inferior thermal stability is very undesirable for processing and application of PLLA. To evaluate the influence of the residual catalysts on thermal stability, the pristine polymer PLLA without any purification after polymerization was subjected to TGA analysis. TGA curves were given in Figure 6 and DTG curves were given in Appendix A. 

Some previous works reported that the residual Sn(Oct)_2_ catalyst in PLLA could deteriorate the thermal stability [37,38,39,40]. The thermal decomposition temperature at 5% mass loss of PLLA obtained by Sn(Oct)_2_ started at around 220 °C, and the thermal decomposition temperature at the maximum mass loss rate appeared at 270 °C. However, the higher residual Sn(Oct)_2_ did not severely deteriorated the thermal stability, evidenced by that these two parameters are closed to the values reported in the very lower residual Sn(Oct)_2_ [37,38,39,40,41]. On the other hand, the thermal decomposition temperatures of most Zn(AA)_2_ catalyzed PLLAs raised about 20~30 °C compared with those of Sn(Oct)_2_ catalyzed PLLA, which clearly endowed more thermal stability of PLLA. In turn, it was attributed to that Sn(Oct)_2_ not only shows high catalytic activity for the polymerization, but also high activity for the polymer decomposition via back-biting as well as transesterification [40].

## 4. Conclusions

A series of zinc salts of different amino acids, including three kinds of amino acids ligands (alanine, phenylalanine, and proline) with l/d-enantiomers and their racemic mixtures, has been successfully synthesized and used as catalysts of l-lactide polymerization. A drawback of partial racemization was observed for zinc amino acids. End-groups and MALDI-TOF MS analysis demonstrated that BnOH is the actual initiator of zinc amino acids catalyzed ROP of l-LA. The results showed that the zinc salts of amino acids first interacted with the BnOH initiator, and the formed intermediate initiated l-LA polymerization. As a result, an activated initiator mechanism proceeded with conventional metal-alkoxides like Sn(Oct)_2_ was proposed. Isothermal crystallization kinetics analysis showed that the residual zinc salts of amino acids exhibited a significant nucleation effect on PLLA, evidenced by the promotion of the crystallization rate, depending on the amino acid ligand and its configuration. PLLA obtained by Zn(l-Pro)_2_ showed the fastest crystallization rate. TGA results showed that the thermal stability of the pristine PLLA was not deteriorated by the residual zinc salts of amino acids. 

A strategy of in situ catalysis and nucleation of biocompatible zinc salts of amino acids towards poly(l-lactide) with enhanced crystallization rate was successfully developed. This one-pot process of catalysis and nucleation is simple and green. It provides an economical, timesaving and effective method in comparison with conventional method of PLLA blended with nucleating agents. However, this work focused on catalysis mechanism and nucleation effect of zinc salts of amino acids at high loading. For the practical applications, our further study will focus on synthesis of high molecular weight and fast crystallization rate PLLA at low concentration of zinc salts of amino acids, and the nucleation efficiency dependence of amino acid ligand and its configuration.

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
