# Peer review of "A Strategy of In Situ Catalysis and Nucleation of Biocompatible Zinc Salts of Amino Acids towards Poly(l-lactide) with Enhanced Crystallization Rate"

_polymers, 2019, doi:10.3390/polym11050790_

Round 1
Reviewer 1 Report
The paper is nicely written and easy to follow. The topic of the manuscript, which is dedicated to the development of biocompatible or nontoxic metal catalysts for the ROP of L-lactide to produce PLLA for biomedical applications, is interesting and important for a wide readership including researchers involved in chemical catalysis and biomedical material research.
The paper is focused on the development of “a novel in situ polymerization of L-lactide in the presence of zinc amino acids to maximize the potential of in situ nucleation” resulting in “a green and low-cost approach of zinc amino acids combining catalysis and nucleation dual roles, for the production of biocompatible PLLA suitable for broad commodity and engineering plastics”. According to the authors, a strategy of in situ catalysis and nucleation of biocompatible zinc amino acids towards poly(L-lactide) with enhanced crystallization rate was successfully developed.
The paper is technically solid. The experimental methodology is described clearly and in detail. The key conclusions of the paper are fully supported by measurements, as evidenced by data shown in Table 2 and Fig.7.
However, a number issues ( a few examples are shown below) related to the use of English should be addressed prior to publication.
Technical issues:
Line 36, delete “more or less”
Line 37, “Even tin(II)” should be replaced with “Even for tin(II)”
Line 38 “catalyst” should be replaced with “catalysts”
Lines 38-39 “at least controversial” r should be replaced with “ at best questionable”
Line 42 “ ideal alternatives” to what?
Line 55 delete “widely”
Line 65 “found to” should be replaced with “found to be”
Author Response
Authors' reply: Thank you very much for your acceptance of our paper. According to your comments, we have corrected the English expression and substantially revised the manuscript by our best efforts.
Reviewer 2 Report
Liang and co-workers report on the synthesis of PLLA by combining the properties of zinc amino-acid complexes to act as catalysts of L-lactide ROP and as nucleating agents to enhance the crystallization rates. Overall, the manuscript is well presented, and the characterization of synthesized polymers was seriously performed by using common techniques. I therefore suggest acceptance after modifications noted below:
1) The reason of why the zinc catalysts allow enhanced crystallization rates should be discussed in more details.
2) P2, line 45: “zinc organocatalyst”!!! Not possible
3) P2, lines 49-50: the sentence “but noting ……. syntheses” is not very clear. The whole paragraph should be re-written to avoid redundancies.
4) Figure 4: it would have been interesting to have data of Sn(Oct)2 catalyst in presence of a common nucleating agent
5) Table 2: k instead of K
Author Response
Authors' reply: Thank you very much for your recommendation of our paper.
(1) A good comment. According to your suggestion, we add a statement that “The nucleation activity may be attributed to the epitaxial nucleation model based on the lattice matching relationship between crystal structures of PLLA crystals and zinc salts of amino acids.” in Line 236 in the revision. And the detailed nucleation mechanism of zinc salts of amino acids will be explained in the next article of this series[36], which will to be submitted.
(2) We have corrected it as Organozinc compounds.
(3) We have re-written it as but the toxicity of these ligands, which required complicated synthetic process, is unknown.
(4) Thank you for your comments. In fact, in reference [33] 33. Im, S H.; Jung, Y.; Kim, S H. In Situ Homologous Polymerization of L-Lactide Having a Stereocomplex Crystal. Macromolecules 2018, 51, 6303-6311. As you said, it did in situ polymerization of L-LA catalyzed by Sn(Oct)2 in presence of PDLA nucleating agent. However, the advantage of this work is to develop a method of dual roles of catalysis and nucleation undertook by one compound.
(5) We have corrected it.
Reviewer 3 Report
The Manuscript polymers-478581 is focused on the synthesis of poly(L-lactide) by new catalysts based on zinc aminoacids. Three aminoacids (alanine, phenylalanine and proline) were used as ligands bowth L/D enantiomers and racemic mixtures. The ability of such zinc salts to act also as nucleating agents was investigated by studying isothermal crystallization of PLLA samples not purified from the catalyst.
General comment
The results are interesting under an applicative point of view since safe zinc salts were used both as catalysts for PLLA synthesis and as nucleating agents for PLLA crystallization.
Specific comments:
1) Line 34: Please delete the word “tremendous”
2) Lines 57-59: I suggest to better specify that a polymer crystallization rate results in a low degree of crystallinity for PLA materials obtained during an injection-molding cycle involving a high cooling rate and a relatively low mold temperature.
3) Lines 147-156: I would add a comment on the fact that proline affects less strongly polymer optical purity than other aminoacids.
4) Lines 191-192. Please check the sentence.
5) Lines 233: Please write the real “n” range, that is 2.35-2.92
6) Lines 235-237: To compare the nucleation ability of the different Zn salts, the authors should use only the t^1/2 column and not the K column. Indeed, K depends on n values (min^-n), and n values are different among the polymers. Therefore, only the values t^1/2 give a right information on the polymer crystallization rate. Otherwise, the authors could consider the possibility to add a column reporting the values of K^1/n.
7) Figure 7: The TGA curve of PLLA obtained by Sn(Oct)2 but purified from the catalyst should be added in order to support the statement “The much poorer thermal stability of PLLA was supposed to be resulted from much more residual Sn(Oct)2” (lines 265-266).
8) Please add just a comment on the potential ability (or not) of residual catalyst to increase polymer degradation by hydrolysis.
Author Response
Authors' reply: Thank you very much for your acceptance of our work.
(1) Maybe the word is too strong, we replaced “tremendous” as “a large number of”.
(2) According to your suggestion, we added the sentence at Line 60.
(3) According to your suggestion, we added the sentence at Line 156.
(4) According to your suggestion, we modified it as “It provided a direct proof that the catalyst first reacted with the initiator BnOH then initiated the monomer L-LA”.
(5) We corrected it.
(6) According to your suggestion, we deleted “k and”.
(7) Thank you for your comment. Some previous works reported that the residual Sn(Oct)2 catalyst in PLLA could deteriorate the thermal stability. However, we cannot provide the experimental result of TGA of the PLLA without the residual Sn(Oct)2 in the limitation of several days. Therefore, we deleted this sentence.
(8) Thank you for your comment. It is a complex question and it is still an open question. We think the promotion or delay of the hydrolysis degradation depends on the nature of the residual catalyst. We cannot add a comment here because of no experimental proof. But we will further investigate this research to explore the hydrolytic degradation behavior of our samples.
This manuscript is a resubmission of an earlier submission. The following is a list of the peer review reports and author responses from that submission.
Round 1
Reviewer 1 Report
The authors presented a ring opening polymerization (ROP) system using zinc amino acids as the catalysts and the nucleating agents. Extensive characterization results, including GPC, DSC and TGA, supported the success of using zinc amino acids as active ROP catalysts to achieve high quality poly(L-lactide) with enhanced crystallization rate. Meanwhile, the authors studied the polymerization mechanism to reveal the role of benzyl alcohol initiator. Overall, the manuscript was well written and the study is comprehensive. I recommend some minor revisions before the manuscript can be published in Polymers.
The authors should provide full characterization data, including 1H NMR, 13C NMR, mass spectroscopy and IR, for the nine zinc amino acids since they were first synthesized.
The authors should provide the integration of peak a and peak d in Figure 1. Discussion of the ratio between the integration of peak a and peak d will also help to gain insights of the real structure of the oligomer.
What is the binding constant/energy between zinc amino acid and BnOH? NMR titration experiment is a good addition in the manuscript.
How does the structure of zinc amino acids affect the crystallization behavior of the polymer? A discussion section might be needed.
Author Response
Please see the Responses to reviewer comments on polymers-457561

Reviewer 2 Report
The authors Liang et al reported the synthesis of zinc amino acids and its utility in the polymerization of L-lactide. The data presented in the article are well organized and very interesting. The manuscript has sound in view of scientific contents and may attract wide scope of scientific readers. Therefore I strongly recommend this manuscript for publication, however, before the publication, following minor points must be addressed.
The authors screened for various Zn catalysts for polymerization of L-lactide and ended with good percentage of product yields. However, the other catalyst Sn(Oct)2 also given 99% of desired polymerized product. What is the rational reason behind the use of Zn catalysts rather than other biocompatible catalysts?
The others have ever tried the same reaction with other lactides?
Overall, the manuscript needs a minor revision.
Author Response

(The authors gave the same response as above.)
